# Constitutive Model of Uniaxial Compressive Behavior for Roller-Compacted Concrete Using Coal Bottom Ash Entirely as Fine Aggregate

Yu Li [1], Li Li [1,2,*] and Vivek Bindiganavile [3]

1 Key Laboratory of Agricultural Soil and Water Engineering in Arid and Semiarid Areas of Ministry of Education, College of Water Resources and Architectural Engineering, Northwest A&F University, Yangling 712100, China; liyu2188@nwafu.edu.cn
2 State Key Laboratory of Green Building Materials, China Building Materials Academy, Beijing 100024, China
3 Department of Civil & Environmental Engineering, University of Alberta, Edmonton, AB T6G 2W2, Canada; vivek@ualberta.ca
* Correspondence: drlili@nwafu.edu.cn

**Abstract:** Coal bottom ash (CBA) is one of the by-products that can be employed as fine aggregate to replace natural sand in concrete. Owing to the very low water demand, roller-compacted concrete (RCC) has the potential to use CBA as fine aggregate at a high proportion. However, little research about RCC using CBA entirely as fine aggregate has been conducted. In this study, the uniaxial compressive strength, deformation, stress–strain curves, and splitting tensile strength of CBA-containing RCC (CBA RCC) were studied to bridge this gap. The compressive strength, elasticity modulus, and splitting tensile strength of all mixtures decreased with increasing CBA content. The relationship between compressive strength and splitting tensile strength of CBA RCC was proposed, which is very close to that recommended by the CEB-FIP code. The uniaxial compressive constitutive model based on the continuum damage theory can well illustrate the stress–strain relationship of CBA RCC. The growth process of damage variable demonstrates the hybrid effect of coarse aggregate, cement, and compacting load on delaying damage under uniaxial compression. The theoretical formula can also accurately illustrate the stress–strain curves of RCC presented in the literature studies.

**Keywords:** constitutive model; uniaxial compression; splitting tensile strength; roller-compacted concrete; coal bottom ash





## 1. Introduction

Coal has been used progressively in the past 20 years as a fuel source around the world [1–7]. Worldwide, 38% of electricity generation and more than 70% of energy required for steelmaking both depend on coal feedstock and burning [8]. However, the increase in coal consumption inevitably produces a large number of by-products, such as fly ash and coal bottom ash (CBA). Fly ash has been adopted widely in concrete [9–11]. CBA is an unfired material of coal burning; nonetheless, current output of CBA in the world is far greater than its utilization. The CBA consists of chromium (Cr), arsenic (As), and mercury (Hg), which pose very serious threat to human health and environment [2,4,12]. Consequently, a large portion of the total CBA is disposed of in landfills and/or ponds throughout the world, leading to expensive disposal costs, loss of resources and energy, and deterioration of the environment. Owing to its huge production and output, CBA disposed of in landfills and/or ponds pollutes groundwater and soil, resulting in very serious environmental problem for a country owning less land or consuming a lot of coal [2,13]. In order to alleviate/eliminate these potential problems, it is essential to develop viable and economical utilization strategies of CBA. For instance, Doğan-Sağlamtimur et al. [3] prepared fired brick with CBA and natural clay. However, in order to protect the environment,

sintered clay bricks have been banned in many countries, such as China. Undoubtedly, the application in the form of concrete is one of the most promising approaches to make use of large quantities of CBA [2,14,15].

Concrete is the most frequently and widely used building material in civil engineering and infrastructures. Therefore, it is very important to study the mechanical properties, durability, and sustainability of concrete [16–22]. Noteworthy, river sand is the most favorable fine aggregate used in concrete production. However, natural river sand is not renewable and will be exhausted gradually. Importantly, the demand to protect the natural environment and to prohibit mining in many countries has further exacerbated the supply problem of river sand. Currently, the scarcity of this basic raw material of concrete is troubling the construction industry. The morphology and particle size distribution of CBA are similar to those of river sand [23]. These characteristics of CBA make it possible to be used as a substitute of river sand as fine aggregate for concrete production. For instance, Jeon et al. [6] used CBA and CaO–CaCl$_2$-activated ground granulated blast furnace slag binder to manufacture artificial fine aggregates by cold-bonded palletization method. CBA could improve the durability and thermal insulation of concrete. Abbas et al. [5] revealed that CBA mitigated alkali–silica reaction expansion of reactive aggregates in concrete. Kim et al. [12] indicated that CBA aggregate did not show an obvious effect on the pore structure of concrete, thus there was no significant difference between chloride penetration depths of the normal concrete and CBA aggregates concrete. Despite this, they concluded that CBA aggregate could obviously decrease the content of chloride diffusion. Savadogo et al. [7] concluded that the partial substitution of CBA powder with 20% mortar did not obviously decrease the durability compared to the control specimen. Yang et al. [24] reported that the increased CBA content lowered the unit weight, thermal conductivity, and compressive strength of concrete. Singh and Siddique [23,25] reported that using CBA as a partial replacement of fine aggregate could endow the specimen with acceptable workability and compressive and splitting strength of concrete. Moreover, using CBA as a partial replacement of fine aggregate lowered the elasticity modulus of concrete, resulting from a less monolithic C–S–H gel structure. However, the use of 100% CBA fine aggregate resulted in a significant decrease in the workability. Kim et al. [2] reported that the use of CBA coarse aggregate showed the greatest effects on the mechanical properties of concrete. However, the rates of decline in mechanical properties resulting from using CBA fine aggregate were controllable by an adjustment of the mix proportion. Noteworthy, the stress–strain constitutive model is the foundation for the analysis of structural behaviors of concrete [26,27]. However, the existing literature studies are mainly focused on the strength of CBA concrete, while the stress–strain constitutive relationship of CBA concrete has not been studied extensively.

Singh and Siddique [23,25] reported that dry CBA decreased the concrete slump significantly. Specifically, Muthusamy et al. [8] reported that when 10–30% CBA was added, the slump flow and passing ability of fresh concrete both decreased [8]. When the fine particles of CBA with size smaller than 1000 μm were added into concrete, the surface area increased and the flowability of concrete decreased. Dry and porous CBA aggregates absorbed the water inside, leading to a decrease in free water and, consequently, decreased the flowability of fresh CBA concrete [25]. Furthermore, the wild shape and rough surface of CBA with meshing performance are also one of the reasons that reduce the flowability of fresh concrete. Specially, replacing 75% of fine aggregate with CBA can reduce slump value by 78% [25]. Therefore, more water is needed in CBA concrete. Nevertheless, excess water has a negative impact on the strength and durability of concrete. This dilemma results in the fact that only a small amount of fine aggregate can be replaced with CBA to balance the workability and hardened properties of concrete [2,23,25]. The slump value of roller-compacted concrete (RCC) is 0 and is rolled to compact its structure [28]. RCC is normally produced with less water, less cement, and more aggregate than normal concrete, usually serving as a preferable structural material for pavement and dam. Owing to its very low water demand, RCC exhibits the potential to replace fine aggregate with CBA in

a high proportion. However, to the best of our knowledge, research about RCC using CBA entirely as fine aggregate has rarely been carried out.

In this research, CBA alone was used totally as fine aggregate to prepare RCC, and the complete stress–strain relationship of CBA-containing RCC (CBA RCC) was comprehensively studied to bridge the gap that exists in literature studies. The influence of CBA content, cement content, coarse aggregate content, and compaction load on the uniaxial compressive and splitting tensile behavior of RCC with CBA fine aggregate were studied herein. The uniaxial compressive and splitting tensile tests of CBA RCC were performed on 108 cylindrical specimens. The compressive strength, splitting tensile strength, compressive tensile strength ratio, and elasticity modulus of RCC were calculated and compared. Moreover, the uniaxial compressive stress–strain relationship was discussed and the constitutive model of CBA RCC was proposed.

## 2. Materials and Methods

### 2.1. Materials

ASTM Type I normal Portland cement (Lehigh Heidelberg Cement Group, Vancouver, Canada) was used. Fine dry CBA is a well-graded fine aggregate, with size gradation within the recommended limits of ASTM C 33 [29]. The maximum size, density, water absorption, and fineness modulus of CBA fine aggregate are 5.00 mm, 1.99 g cm$^{-3}$, 10.0%, and 3.01, respectively. The results of an X-ray fluorescence test presented that CBA mainly consists of $SiO_2$ (49.9%), $Al_2O_3$ (13.1%), and $Fe_2O_3$ (23.0%), accompanied with little CaO (0.80%), MgO (0.38%), and $SO_3$ (0.26%). The grain size distribution of the graded coarse aggregate meets the standard of the permissible size ranges for stone No. 67 by ASTM C 33 [29]. Tap water was used for all the experiments.

### 2.2. Mixture Proportion and Specimen Product

There were 36 different groups of test specimen, and 108 test specimens were prepared in total. Nine concrete mixtures each involving four batches were prepared. According to the literature [28,30], the water/solid ratio of RCC generally ranges from 4.5 to 6.6%. In this research, the water/solid ratio was fixed at 7%, because of the high porosity of CBA aggregate. Mixture proportions of three levels of cement content ranging from 9, 12 to 15% (by mass of the total dry solids) and three levels of coarse aggregate content from 50, 55 to 60% (by mass of the total dry solids) were used. Thus, the corresponding CBA content varied from 25 to 41%. Mixture proportions are presented in Table 1. Yang et al. [24] used 24% CBA fine aggregate (by mass of the total dry solids) to prepare concrete. Kim et al. [2] used 30 and 32% CBA fine aggregate (by mass of the total dry solids) to prepare concrete. As a result, more CBA can be used in RCC than in normal concrete. CBA content varied from 25 to 41% in this research, which is reasonable.

**Table 1.** Mixture constituents and proportions.

| Mix No. | Cement | Bottom Ash | Coarse Aggregate | Cement | Bottom Ash | Coarse Aggregate |
|---------|--------|-----------|------------------|--------|-----------|------------------|
| | % | | | kg m$^{-3}$ | | |
| S9-1 | 9 | 41 | 50 | 187 | 853 | 1041 |
| S9-2 | 9 | 36 | 55 | 191 | 764 | 1160 |
| S9-3 | 9 | 31 | 60 | 192 | 662 | 1281 |
| S12-1 | 12 | 38 | 50 | 254 | 803 | 1057 |
| S12-2 | 12 | 33 | 55 | 256 | 704 | 1173 |
| S12-3 | 12 | 28 | 60 | 260 | 608 | 1302 |
| S15-1 | 15 | 35 | 50 | 328 | 765 | 1093 |
| S15-2 | 15 | 30 | 55 | 332 | 664 | 1217 |
| S15-3 | 15 | 25 | 60 | 335 | 558 | 1339 |

The following designation (in Table 2) was used to identify the testing specimens. Testing specimen designation is Sa-b-c-d (or Gm-n), where S: specimen; a: cement content (%), in this experimental program, there were three levels of cement content, which were 9, 12, and 15%; b: number of the different groups of the same a; c: number of the different groups of the same b; d: number of the different specimens of the same c; G: group; m: number of different groups; n: number of different groups of the same m.

**Table 2.** Group and specimen number.

| G1-1 | G1-2 | G1-3 | G1-4 |
|------|------|------|------|
| S9-1-1-1 | S9-1-2-1 | S9-1-3-1 | S9-1-4-1 |
| S9-1-1-2 | S9-1-2-2 | S9-1-3-2 | S9-1-4-2 |
| S9-1-1-3 | S9-1-2-3 | S9-1-3-3 | S9-1-4-3 |

As shown in Table 2, G1-1 indicates the first group of the first section of 9% cement content (by the mass of the total dry solids); S9-1-1-1: 1# specimen of the first group of the first group of 9% cement content (by the mass of the total dry solids); G1-1 and G1-2: surcharge is 20 lb (9.305 kg); G1-3 and G1-4: surcharge is 40 lb (18.610 kg); G1-1 for testing compressive strength at 28 days; G1-2 for testing splitting tensile strength at 28 days; G1-3 for testing compressive strength at 28 days; G1-4 for testing splitting tensile strength at 28 days.t

Specimens were prepared under a normal temperature condition (about 25 °C) at Annie Hole/Ted Hole Memorial Laboratory in the University of Alberta. Red Lion MIXER (Model: RLX-6, Type: B, Ken's Distributing Company, Denver, CO, USA) was used in mixing RCC containing CBA. A single-use plastic, cylindrical mold with 4 in. (100 mm) in diameter and 8 in. (200 mm) in height was used to fill with concrete. A VIBRO-PLUS Vibrating Table (Type: VBR, Martin Vibration Systems & Solutions, Marine City, MI, USA) was used to consolidate the test specimens of RCC containing CBA. Vibrating time was $60 \pm 20$ s. Trowels, a square-ended shovel, hand scoops, a steel trowel, and tamping rod were used in this experimental program. After casting, the molded specimens were covered with a plastic lid and left in the casting room for 24 h. They were then demolded and placed in a standard humidity room to be cured for 28 days until testing.

### 2.3. Test Setup

Before conducting the compressive strength test and obtaining the stress–strain curve, the specimens were capped on one side at the capping station using sulfur at 130 °C, and then, they were put into the humidity room to be cured again until testing at 28 days. The other side of the specimens was placed on a poly wood pad when testing.

The tests of compressive strength and stress–strain curve of the specimens were carried out at the I. F. Morrison Structural Engineering Laboratory using the 2600 kN load capacity of MTS servo hydraulic closed loop testing machine (MTS Systems Corporation, Toronto, Canada), applying a monotonically growing displacement loading with a constant speed of 0.01 mm s$^{-1}$ [31]. A compressometer (MTS Systems Corporation, Toronto, Canada) with two linear variable differential transducers was fixed to the specimen to measure its deformation. The splitting tensile strength test was accomplished at Annie Hole Memorial Laboratory using TINIUS OLSEN testing machine (Tinius Olsen Inc., Horsham, PA, USA).

### 3. Results and Discussion
### 3.1. Compressive Strength

Figure 1 exhibits that the compressive strength becomes improved with increasing cement content. Moreover, the influence of varying compaction load on compressive strength of the same mixture did not show a significant difference except from the group (G8-3: S15-2-3-1 and S15-2-3-2) with 15% cement content, 55% coarse aggregate content, and 40 lb compaction load. There were only six fine vertical cracks, which were formed by running through the cap on one side of S15-2-3-1, and small buckle was formed on

one side of S15-2-3-2. For the mixture containing 55% coarse aggregate, the compressive strength increased by 129.63% (with surcharge 20 lb) and 110.73% (with surcharge 40 lb) when cement content was elevated from 9 to 12%. An increase of 72.10% (with surcharge 20 lb) and 170.23% (with surcharge 40 lb) in compressive strength occurred for the same coarse aggregate content when cement content was increased from 12 to 15%. The increase in cement content resulted in a higher availability of cement paste to bond the aggregates. This produced better interlocking behavior and improved bond strength of solid particles. The different surcharge showed the similar influence on compressive strength of RCC specimens containing CBA at 28 days.

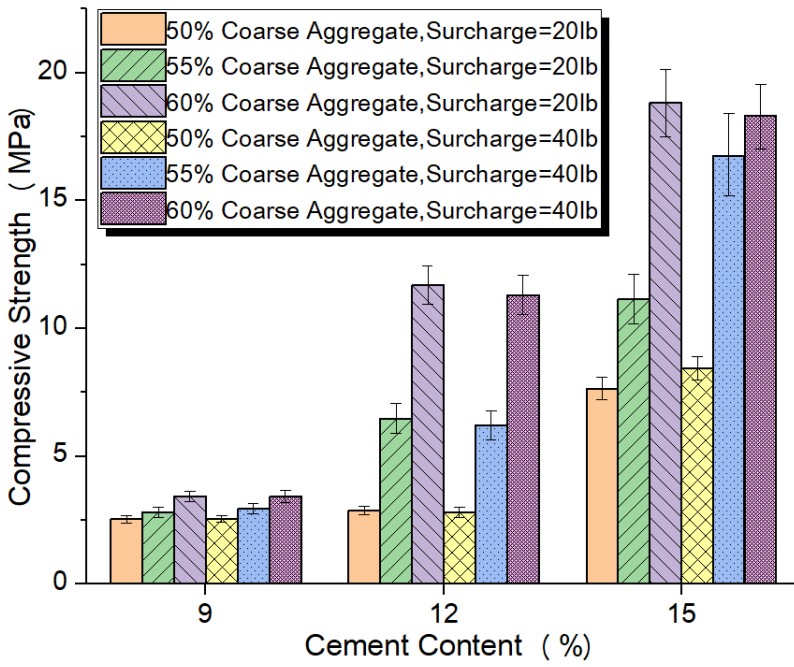

**Figure 1.** Influence of compaction load and cement content on compressive strength.

Figure 2 demonstrates that the compressive strength decreases with the increase in CBA content (by mass of total dry solids), due to the high porosity of CBA aggregate. In general, for 9 and 15% cement content groups, the compressive strengths of the same mixture specimens improved with the increase in the compaction load from 20 to 40 lb. This results from a more compact structure due to higher compaction load. For 30% CBA content in the 15% cement content group, the 28-day compressive strength increased by 50.67% when surcharge was increased from 20 to 40 lb. However, for 36% CBA content in the 9% cement content group, the compressive strength exhibited only 4.7% improvement when the compaction load was increased from 20 to 40 lb. Nonetheless, for the 12% cement content group, the compressive strength decreased slightly with the increase in surcharge from 20 to 40 lb.

Figure 3 illustrates that for the 9% cement content with 20 and 40 lb surcharge, the 28-day compressive strength increased by 11.20 and 15.83%, respectively, when the coarse aggregate content increased from 50 to 55%. The compressive strength was improved by 21.78 and 16.05%, respectively, when the coarse aggregate content increased from 55 to 60%. For the 12% cement content with 20 and 40 lb surcharge, the compressive strength increased by 124.15 and 121.23%, respectively, when the coarse aggregate content was increased from 50 to 55%. The compressive strength improved by 80.56 and 81.98%, respectively, when the coarse aggregate content was increased from 55 to 60%. For the 15% cement content with 20 and 40 lb surcharge, the 28-day compressive strength increased by 45.67 and 97.62%, respectively, with the increase in the coarse aggregate content from 50 to 55%. The 28-day compressive strength was found to improve by 68.91 and 8.97%, respectively, with the increase in the coarse aggregate content from 55 to 60%. The strength improvement, with

increase in coarse aggregate content, is a direct consequence of the decreased high porosity of CBA aggregate.

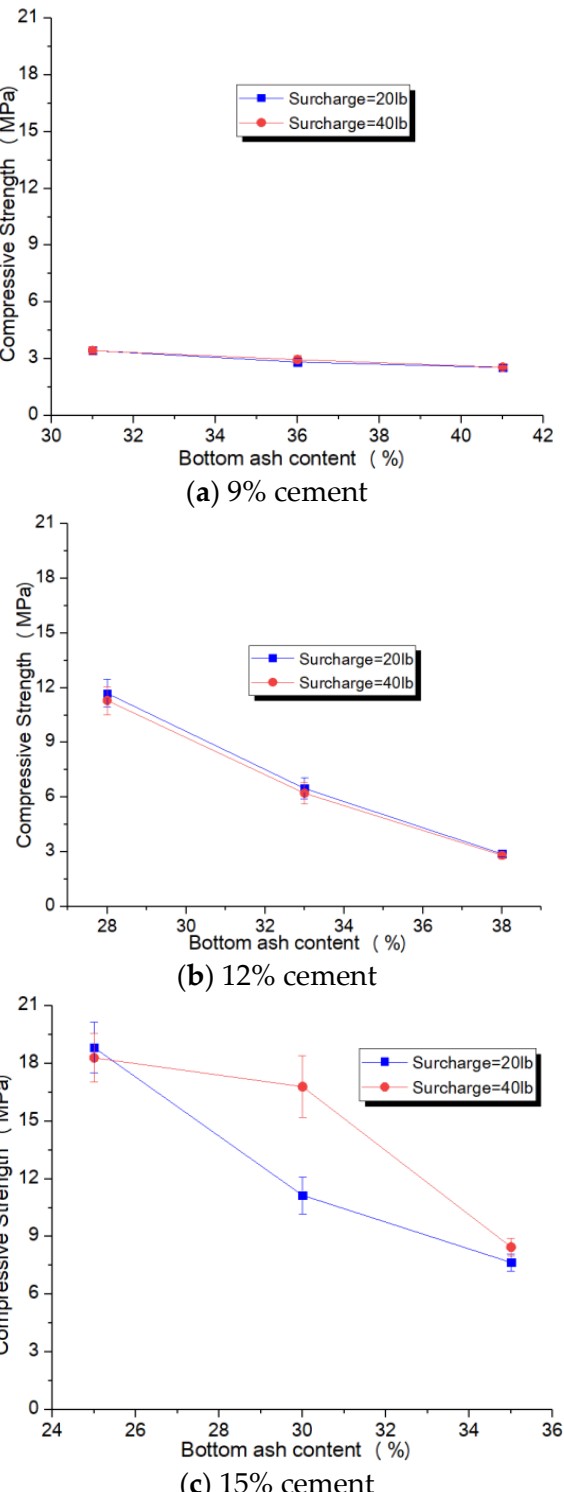

**Figure 2.** Influence of bottom ash content on compressive strength in three specimens with different cement contents.

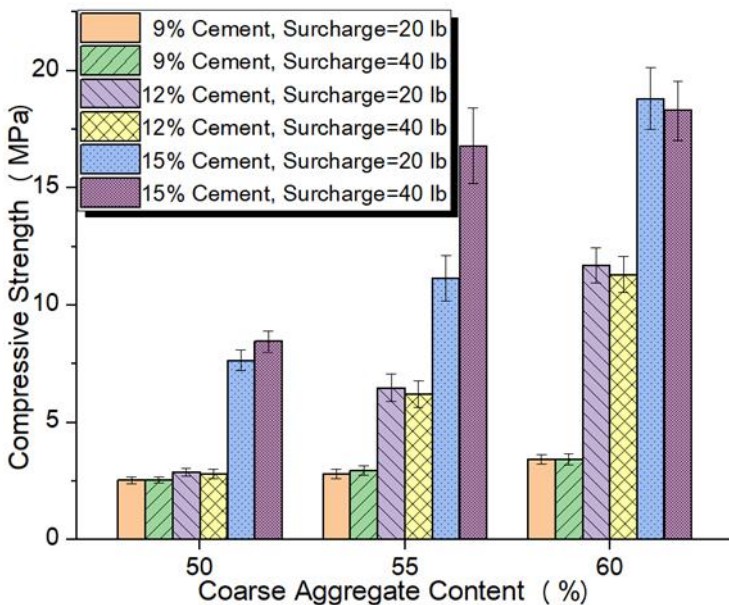

**Figure 3.** Influence of coarse aggregate content on compressive strength.

### 3.2. Splitting Tensile Strength

Figure 4 presents that similar to the compressive strength, the splitting tensile strength also becomes improved with the increase in cement content. For the specimens with 55% coarse aggregate, the 28-day splitting tensile strength was increased by 128.21% (20 lb surcharge) and 115.18% (40 lb surcharge) with the increase in the cement content from 9 to 12%, and by 93.82% (20 lb surcharge) and 88.83% (40 lb surcharge) when the cement content was increased from 12 to 15%. When specimen consisted of 60% coarse aggregate, the improvement in splitting tensile strength was 214.68% (20 lb surcharge) and 206.00% (40 lb surcharge), respectively. Notably, the overall rate of growth in splitting tensile strength was slightly greater in the group containing 55 or 60% coarse aggregate content with 20 lb surcharge than those with 40 lb surcharge. This indicates that the varying compaction load did not exhibit a distinct effect on splitting tensile strength, similar to compressive strength.

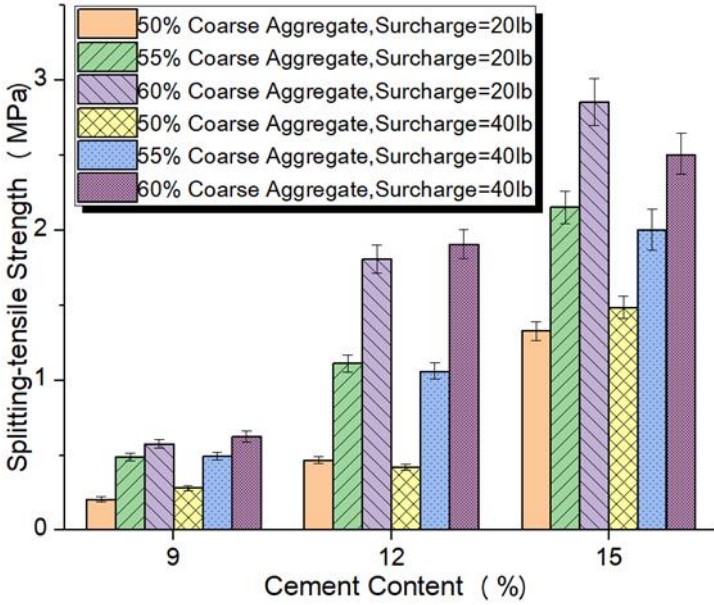

**Figure 4.** Influence of compaction load and cement contents on splitting ten-sile strength.

Figure 5 presents the effect of different CBA contents on splitting tensile strength of CBA RCC with three different cement contents. The trends are similar to those of compressive strength, i.e., the splitting tensile strength decreased with increasing CBA content (by mass of total dry solids). On the whole, for the 9% cement content group, the splitting tensile strength was found to improve with the increase in the compaction load from 20 to 40 lb. However, for the 12% cement content group, the splitting tensile strength decreased slightly with the increase in surcharge from 20 to 40 lb. Moreover, for the 15% cement content group, the splitting tensile strength decreased significantly with the increase in surcharge from 20 to 40 lb.

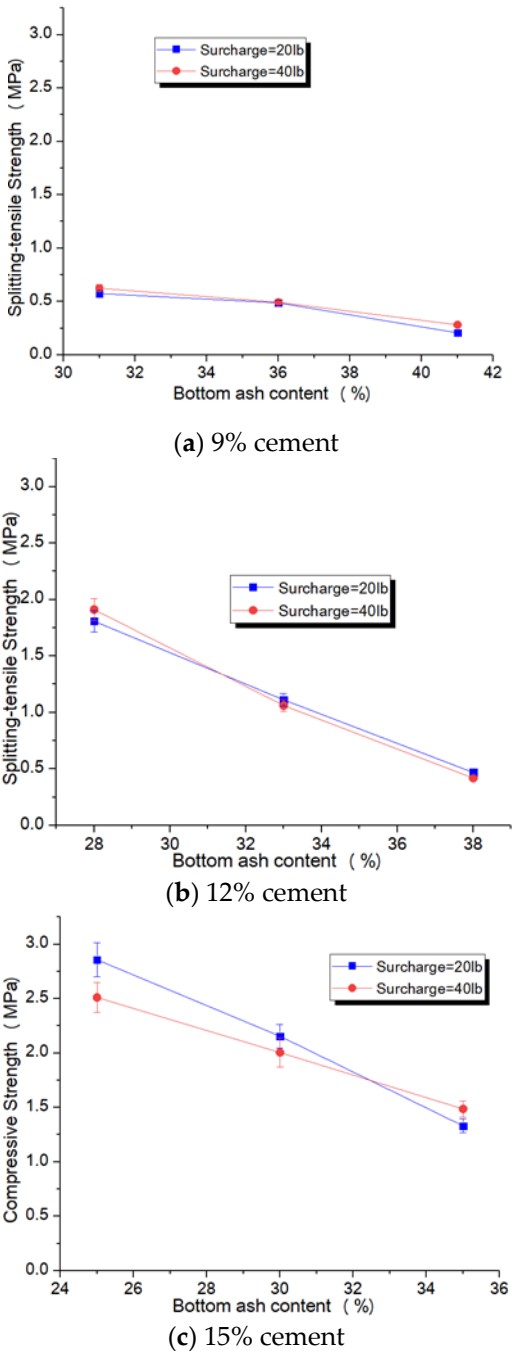

(**a**) 9% cement

(**b**) 12% cement

(**c**) 15% cement

**Figure 5.** Influence of bottom ash content on splitting tensile strength of cement with three different contents.

Table 3 summarizes that for Group 1 with smaller cement content (9% by mass of the total dry solids), in general, the rate of reduction in splitting tensile strength was greater than that of compressive strength when CBA content was increased from 31 to 36 and finally to 41%. Irrespective of the compaction load applied, i.e., 20 or 40 lb, the same trend was obtained. Variation of CBA content showed more influence on splitting tensile strength than on compressive strength in the smaller cement content. This is attributed to more sensitivity of splitting tensile strength to porosity than that of compressive strength. For Group 2 with 12% cement content, the rates of reduction in splitting tensile strength and compressive strength present the same level of reduction, which indicates that varying CBA content showed similar influence on splitting tensile strength and compressive strength in the group with 12% cement content. For Group 3 with 15% cement content, the rate of reduction between splitting tensile strength and compressive strength presents more complex results. For 40 lb compaction load, the rate of reduction in splitting tensile strength is greater than that of compressive strength when CBA content was increased from 25 to 30%. However, for the same level of compaction load, the rate of reduction in compressive strength was greater than that of splitting tensile strength when CBA content was increased from 30 to 35%.

**Table 3.** Influencing rate of reduction of varying bottom ash content on splitting tensile strength and compressive strength.

| The Changing Range of CBA | | Reduction Rate of Splitting Tensile Strength (%) | | Reduction Rate of Compressive Strength (%) | |
|---|---|---|---|---|---|
| | | Surcharge = 20 lb | Surcharge = 40 lb | Surcharge = 20 lb | Surcharge = 40 lb |
| 9% cement content | CBA from 31 to 36% | 15.22 | 21.00 | 18.87 | 13.82 |
| | CBA from 36 to 41% | 57.60 | 43.04 | 10.07 | 13.67 |
| 12% cement content | CBA from 28 to 33% | 38.51 | 44.45 | 44.62 | 45.00 |
| | CBA from 33 to 38% | 57.87 | 54.41 | 55.39 | 54.83 |
| 15% cement content | CBA from 25 to 30% | 24.59 | 19.86 | 40.80 | 8.23 |
| | CBA from 30 to 35% | 38.26 | 25.86 | 31.36 | 49.68 |

Figure 6 demonstrates that for the three levels of cement content, namely, 9, 12, and 15%, the influence graphs of coarse aggregate content on the 28-day splitting tensile strength present a similar changing trend to the compressive strength.

### 3.3. Relationship between Compressive Strength and Splitting Tensile Strength

The tensile strength to compressive strength (T/C) ratio is an effective assessment indicator of toughness for concrete. In general, a high T/C ratio generally indicates a good toughness [32,33]. Figure 7 presents that for the mixtures containing 50% coarse aggregate with 20 and 40 lb surcharge, the T/C ratio increased with the increase in cement content, and the surcharge resulted in the decrease in the increasing rate. However, for the mixtures containing 60% coarse aggregate with 20 and 40 lb surcharge, the T/C ratio decreased with the increase in cement content, and the surcharge increased the decreasing rate. Moreover, the dispersion of the T/C ratio data depended on the cement content. Specifically, the coarse content and surcharge exhibited minimal influence on the mixtures containing 12% cement, while they showed significantly more influence on the mixtures containing 9 and 15% cement, respectively.

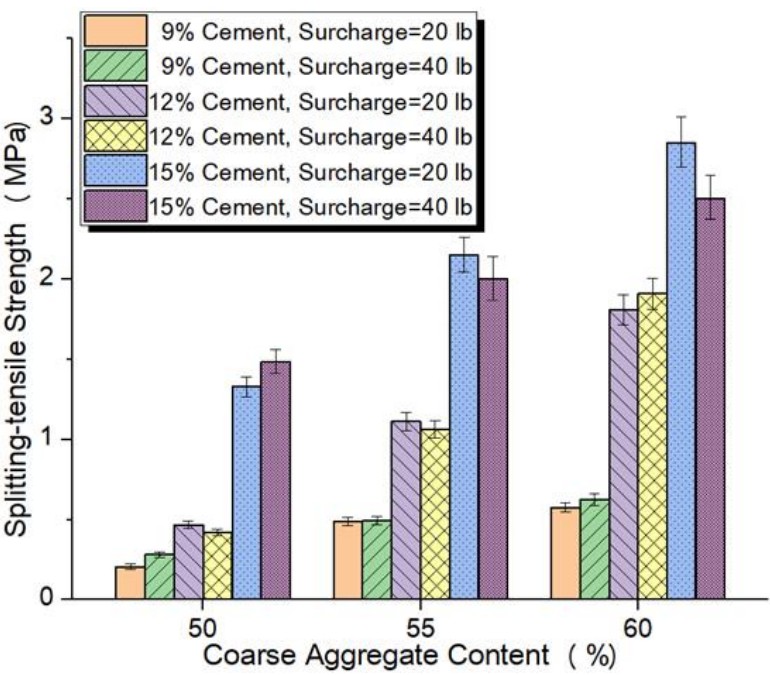

**Figure 6.** Influence of coarse aggregate content on splitting tensile strength.

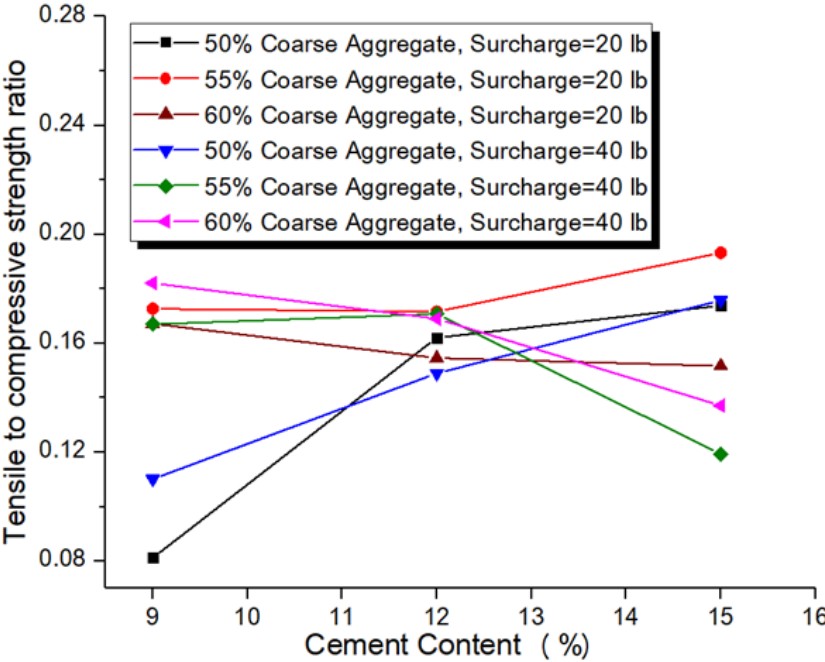

**Figure 7.** Influence of cement contents on tensile–compressive strength ratio.

Figure 8 presents the effect of different CBA contents on T/C ratio of RCC with three different cement contents. For the RCC with 40 lb surcharge, it is not easy to indicate the trend of T/C ratio. Nonetheless, for the RCC subjected to 20 lb surcharge, the T/C ratio increased first and then decreased regardless of the cement content. When CBA content ranged from 30 to 36%, the CBA RCC reached higher toughness with 20 lb surcharge.

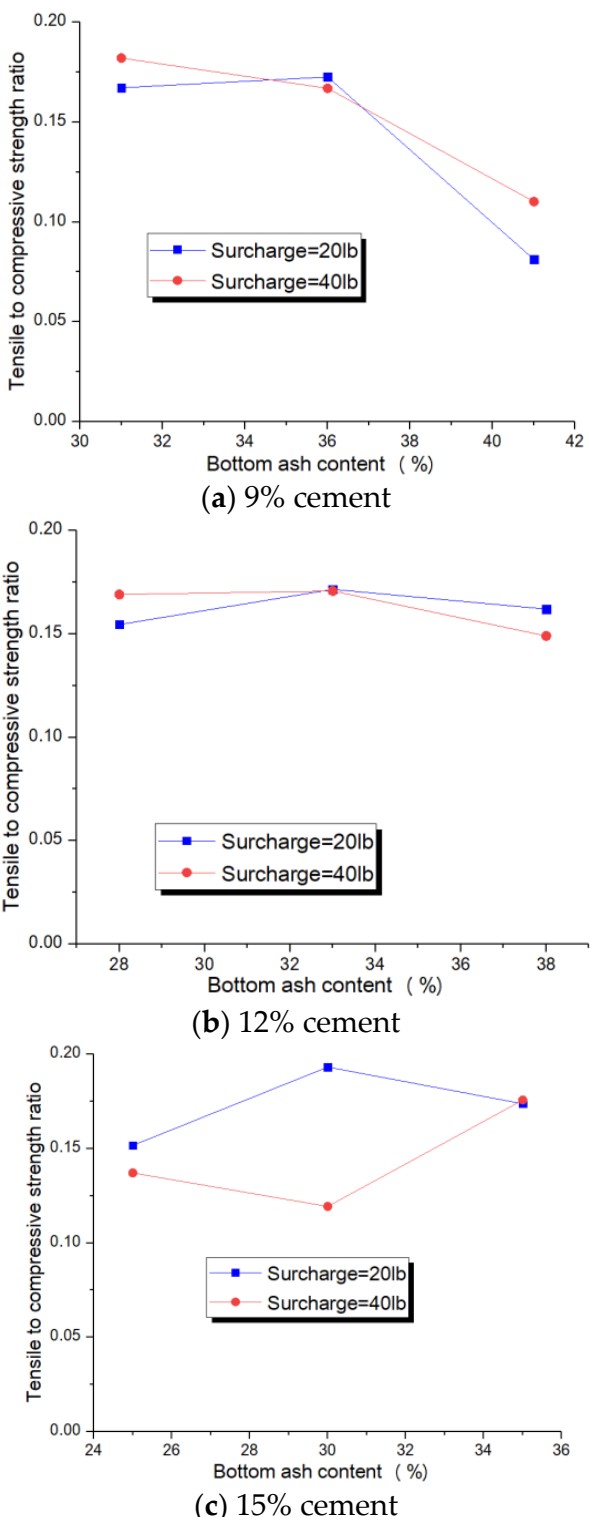

**Figure 8.** Influence of CBA content on tensile–compressive strength ratio in cement with three different contents.

Figure 9 illustrates that for the mixtures containing 9 and 12% cement, the T/C ratio increases with the increase in coarse aggregate content and the surcharge leads to the decrease in the increasing rate, in general. Nonetheless, for the mixtures containing 15% cement, the variation trend of T/C ratio was not clear.

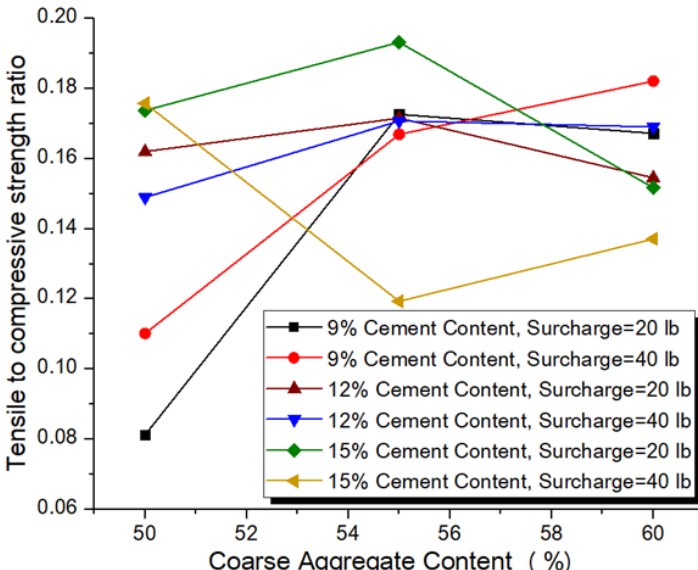

**Figure 9.** Influence of coarse aggregate content on tensile–compressive strength ratio.

Figure 10 reveals the relationship between splitting tensile strength and compressive strength of CBA RCC. The equation displaying the relationship between compressive strength and splitting tensile strength together with the coefficients of determination $R^2$ derived from the test results of the present study is presented as follows:

$$f_t = 0.2988 f_c^{0.7433}, R^2 = 0.818 \tag{1}$$

where $f_t$ = splitting tensile strength in MPa, and $f_c$ = compressive strength of cylinder in MPa. The higher value of coefficient of determination $R^2$ shows a good relationship between regression curve and data points. The relation is very close to the relationship $f_t = 0.3 f_c^{(2/3)}$, recommended by CEB-FIP model code for concrete structures (1990) [34]. Figure 10 displays that the curve obtained in this study is steeper than the CEB-FIP curve [34], indicating that the T/C ratio of CBA RCC in this study is higher than that of normal concrete. The higher toughness of CBA RCC results from the highly porous CBA aggregate.

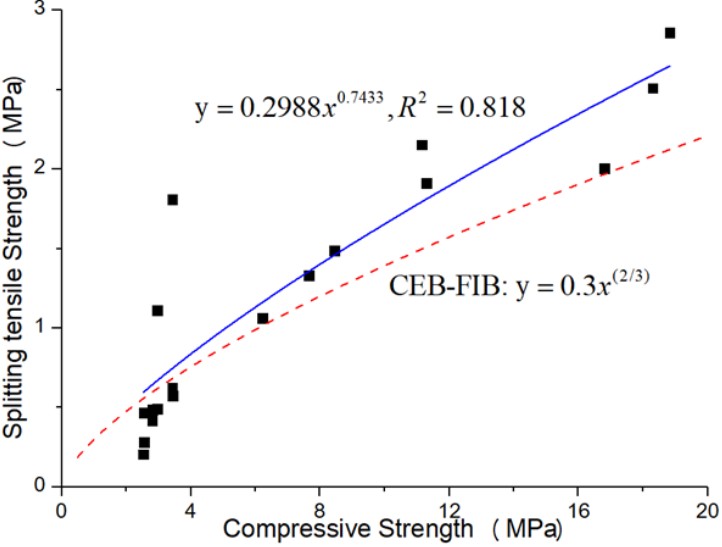

**Figure 10.** Relationship between compressive strength and splitting tensile strength.

### 3.4. Elastic Modulus

Modulus of elasticity is generally employed to illustrate the longitudinal stiffness of material at elastic phase. The elastic modulus is defined as the slope of the uniaxial compressive stress–strain curve between 40% peak stress and a stress corresponding to a strain of 0.00005, according to ASTM C469. The elastic modulus of CBA RCC with various cement content is shown in Figure 11, revealing that the elastic modulus increases significantly with increasing cement content. In general, the surcharge elevated the increasing rate except for the RCC with 50% coarse aggregate. The reason for the increased stiffness of RCC was that the cement increased the bond between solids, and the compacting load further elevated this trend. Moreover, the dispersion of the elastic modulus data depends on the cement content. Specifically, the coarse content and surcharge exhibited minimal influence on the mixtures containing 9% cement, while they showed significantly more influence on the mixtures containing 15% cement.

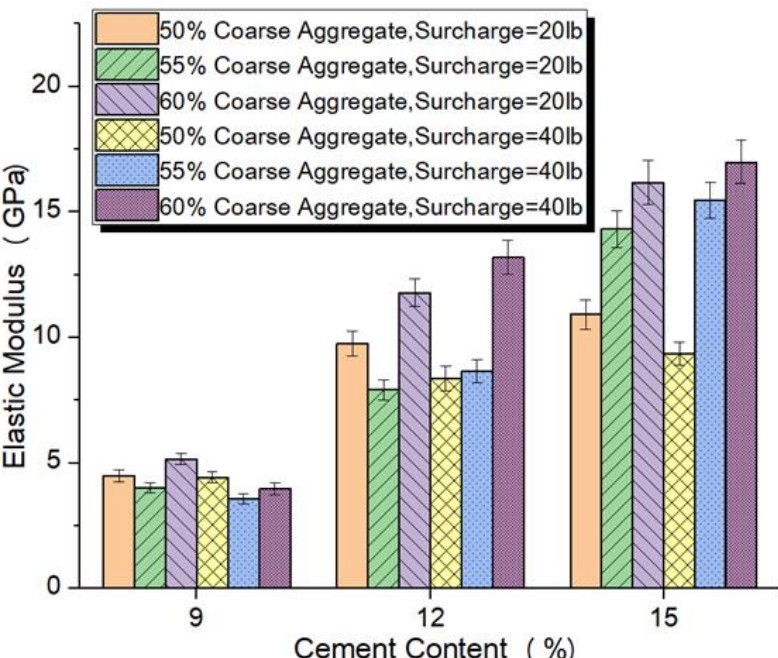

**Figure 11.** Influence of cement contents on elastic modulus.

Figure 12 presents the effect of different CBA contents on elastic modulus of RCC with three different cement contents. Clearly, the elastic modulus decreased with CBA content regardless of the cement content. When CBA content was about 25%, the elastic modulus of CBA RCC could reach higher than 16 GPa. The compacting load had less effect on the elastic modulus of CBA RCC in this research.

Figure 13 presents that the elastic modulus increases generally with increasing coarse aggregate content. For the mixtures containing 15% cement, the increasing rate of elastic modulus was the highest, the mixtures containing 12% cement acquired the second place, while the mixtures containing 9% cement presented the lowest increasing rate. This result is attributed to the fact that the interlock between coarse aggregate could increase the elastic modulus. Moreover, the bonding from cement could increase the interlock effect of coarse aggregate.

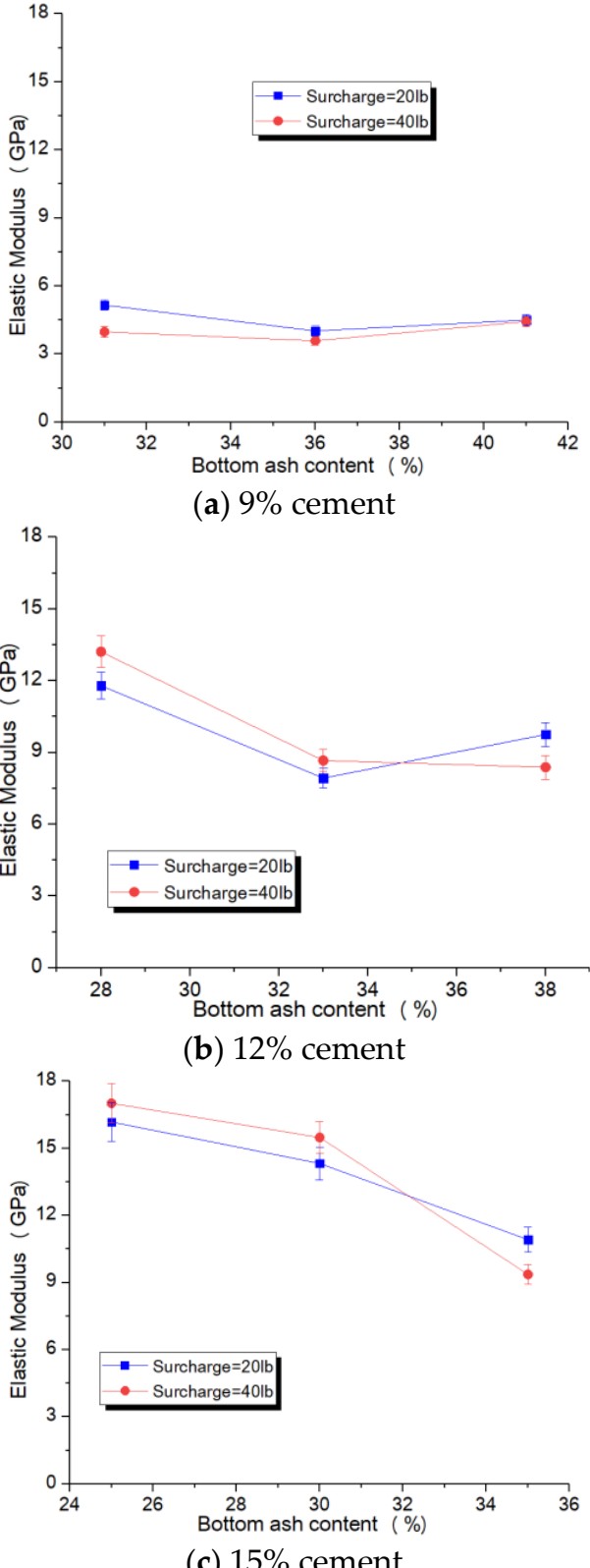

**Figure 12.** Influence of bottom ash content on elastic modulus of RCC with three different cement contents.

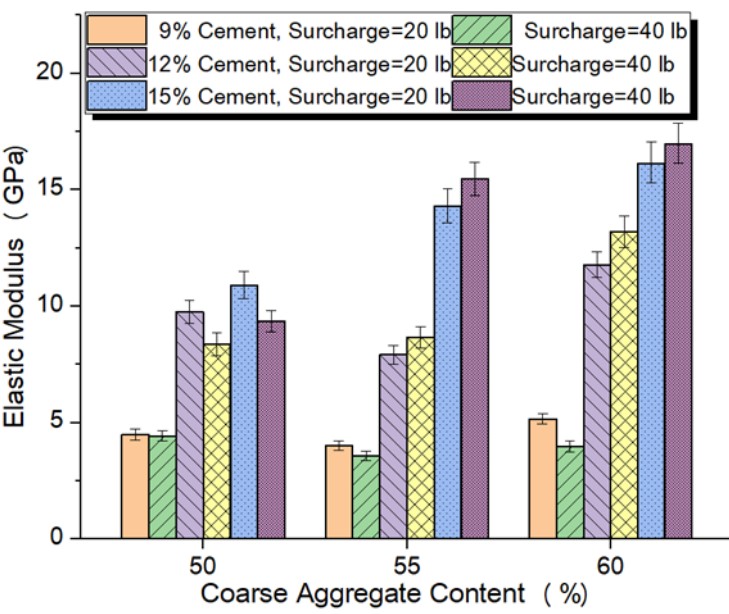

**Figure 13.** Influence of coarse aggregate content on elastic modulus.

*3.5. Uniaxial Compressive Stress–Strain Relationship*

The typical uniaxial compressive stress–strain curves of CBA RCC with 9% cement content and 50% coarse aggregate are shown in Figure 14a,b, respectively. The climbing phase of stress–strain curves of CBA RCC contains elastic stage and elastic–plastic stage (labeled as OB) [27]. Figure 14 clearly presents the existence of a linear relationship between stress and strain of RCC in the OA stage. The AB stage shown in Figure 14 indicates the stable crack propagation phase, in which the CBA fine aggregate and coarse aggregate can restrict the development of cracking and transfer crack tip stress. The BC stage indicates the unstable crack propagation phase, in which the coarse aggregates can interlock with each other and bridge cracks and can gradually be pulled out. After point C, the coarse aggregates cannot interlock each other, the strain increases rapidly, and then, the specimens suddenly break into pieces exhibiting a typical brittleness material characteristic.

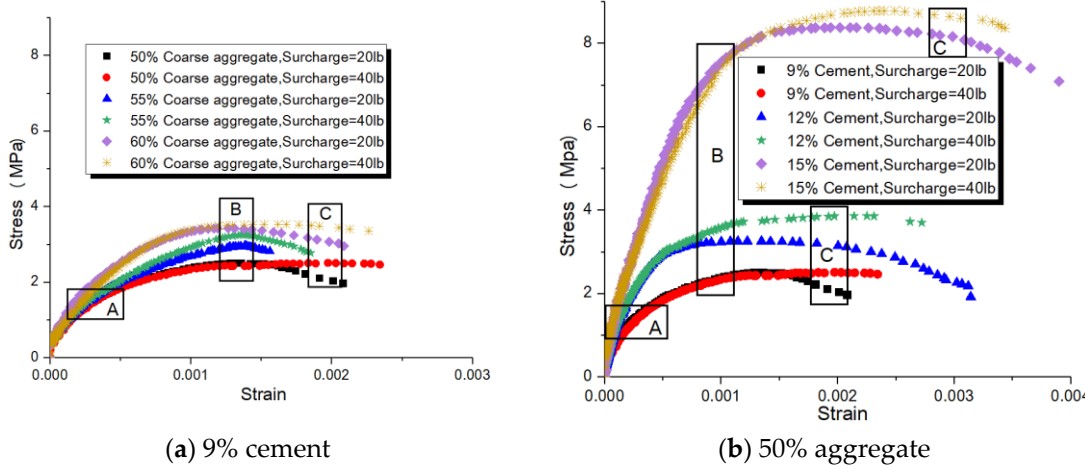

**(a)** 9% cement  **(b)** 50% aggregate

**Figure 14.** Typical uniaxial compressive stress–strain curves of CBA RCC with 9% cement content and 50% coarse aggregate.

Figure 14a clearly demonstrates that the differences of stress–strain curves between specimens with various coarse aggregate are minimal in OA stage. Further, the differences of stress–strain curves between specimens with various coarse aggregates are widened gradually in AB stage. The differences of stress–strain curves between specimens with

various coarse aggregates are the greatest in BC stage. This indicates that the coarse aggregates can play an interlocking effect more efficiently in the elastic–plastic stage. Figure 14b exhibits that the differences of stress–strain curves between specimens with various cement content present similar trend; however, the differences of stress–strain curves between specimens with various cement content are much more significant than those with various coarse content. This demonstrates that the chemical bonding of solids in CBA RCC exhibits controlling effects on the crack propagation process.

Figure 14 illustrates that the compacting load has little effect on the stress–strain curve in the OA and AB stages; however, it exhibits a significant effect on the stress–strain curve in the BC stage. This indicates that the compacting load plays its role mainly in the unstable crack propagation phase, leading to the improvement in the interlocking and pulling out effect of coarse aggregate. Although the compacting load has little effect on the initial elastic modulus of CBA RCC, it can improve the deformation ability and toughness of CBA RCC.

*3.6. Uniaxial Compressive Constitutive Model and Damage Variable*

The influence of raw material on the mechanical properties between concretes primarily depends on initial the crack quantity and crack propagation behavior under loading. The local stress concentration behavior occurs in concrete under the increase in compressive load, then the crack generates in concrete and the damage occurs. The damage variable $D$ is employed to illustrate the appearance and development of cracking process [35]; $D = 0$ represents no damage, and $D = 1$ represents total damage. Moreover, it is assumed that the damage results from the failure of local microregion. If the area of failure microregion $B$ is under certain compressive loading, the damage variable $D$ can be defined as the ratio between the failure area $B$ and total area $A$ of microregion as follows:

$$D = B/A \tag{2}$$

The following damage constitutive model is deduced depending on the continuum damage mechanics [36]:

$$\sigma = \varepsilon E(1 - D) \tag{3}$$

As a random variable, the damage variable of RCC is influenced by several parameters such as porosities, initial cracks, and hydration products. The parameters are independent of each other and random variables conforming to statistical rules. Therefore, the damage variable of RCC can be illustrated by using the statistical distribution. The literature study presents that the damage variable of concrete conforms to Weibull distribution [37]. Assuming that the damage variable of RCC follows Weibull distribution, the probability density function is described as follows:

$$D = 1 - \exp[-(\varepsilon/\eta)^m] \tag{4}$$

where $\varepsilon$ is strain of RCC, and $m$ and $\eta$ are parameters characterizing the shape and size factor, respectively. Then, the following damage constitutive model is deduced:

$$\sigma = \varepsilon E(1 - D) = \varepsilon E \exp[-(\varepsilon/\eta)^m] \tag{5}$$

The slope at the peak stress point is 0, then

$$\frac{\mathrm{d}\sigma}{\mathrm{d}\varepsilon} = [1 - m(\varepsilon/\eta)^m]E \exp[-(\varepsilon/\eta)^m] = 0 \tag{6}$$

On account of $E \neq 0$ and $\exp[-(\varepsilon/\eta)^m] \neq 0$, then

$$1 - m(\varepsilon_{\mathrm{pk}}/\eta)^m = 0 \tag{7}$$

That is,

$$\eta = \varepsilon_{pk}/(1/m)^{\frac{1}{m}} \tag{8}$$

Combining Formulas (5) and (8), the parameters $m$ can be calculated as follows:

$$m = 1/\ln(E\varepsilon_{pk}/\sigma_{pk}) \tag{9}$$

Formula (9) indicates that the toughness of RCC decreases with the increase in $m$. Therefore, $m$ illustrates the concentration grade of microregion strength. Moreover, $\eta$ describes the macroscopic statistical strength. The constitutive model parameters of CBA RCC calculated by using Formulas (8) and (9) are presented in Table 4.

**Table 4.** Constitutive model parameters of CBA RCC.

| | Group | Peak Strain | Peak Stress | Elastic Modulus | $m$ | $\eta$ |
|---|---|---|---|---|---|---|
| 50% Coarse aggregate | 9% Cement, Surcharge = 20 lb | 0.00132 | 2.51506 | 4.5017 | 1.163074 | 0.001503 |
| | 9% Cement, Surcharge = 40 lb | 0.00197 | 2.51739 | 4.4308 | 0.804252 | 0.001503 |
| | 12% Cement, Surcharge = 20 lb | 0.00122 | 3.26293 | 9.7559 | 0.772739 | 0.000874 |
| | 12% Cement, Surcharge = 40 lb | 0.00214 | 3.86745 | 8.382 | 0.651764 | 0.00111 |
| | 15% Cement, Surcharge = 20 lb | 0.00211 | 8.3839 | 10.917 | 0.989417 | 0.002087 |
| | 15% Cement, Surcharge = 40 lb | 0.00231 | 8.78415 | 9.3641 | 1.109654 | 0.002537 |
| 55% Coarse aggregate | 9% Cement, Surcharge = 20 lb | 0.00138 | 2.97761 | 4.0173 | 1.608823 | 0.001855 |
| | 9% Cement, Surcharge = 40 lb | 0.00138 | 3.27761 | 3.5874 | 2.42485 | 0.001988 |
| | 12% Cement, Surcharge = 20 lb | 0.00230 | 6.78966 | 7.9287 | 1.012148 | 0.002328 |
| | 12% Cement, Surcharge = 40 lb | 0.00137 | 6.38761 | 8.6624 | 1.614355 | 0.001843 |
| | 15% Cement, Surcharge = 20 lb | 0.00329 | 10.65057 | 14.325 | 0.672368 | 0.001823 |
| | 15% Cement, Surcharge = 40 lb | 0.00197 | 17.78611 | 15.484 | 1.853833 | 0.002748 |
| 60% Coarse aggregate | 9% Cement, Surcharge = 20 lb | 0.0013 | 3.43541 | 5.1618 | 1.493622 | 0.001701 |
| | 9% Cement, Surcharge = 40 lb | 0.0015 | 3.53818 | 3.9754 | 1.91579 | 0.002106 |
| | 12% Cement, Surcharge = 20 lb | 0.0036 | 11.91005 | 11.791 | 0.786852 | 0.002655 |
| | 12% Cement, Surcharge = 40 lb | 0.0018 | 12.78777 | 13.212 | 1.611804 | 0.00242 |
| | 15% Cement, Surcharge = 20 lb | 0.0021 | 19.24688 | 16.175 | 1.760392 | 0.002896 |
| | 15% Cement, Surcharge = 40 lb | 0.0023 | 25.7535 | 17.012 | 2.390871 | 0.003312 |

Figure 15a presents the comparative analysis of theoretical and experimental uniaxial compressive stress–strain curves of CBA RCC with 9% cement. The theoretical to experimental stress ratio of CBA RCC with 9% cement is shown in Figure 15b. Figure 15 shows that the theoretical formula can accurately illustrate the stress–strain curves of CBA RCC, and most of the theoretical to experimental stress ratios range from 0.9 to 1.1. The intro-

duction of 60% coarse aggregate decreases the correlation degree between theoretical and experimental uniaxial compressive stress–strain curves before peak stress, but improves it after peak stress.

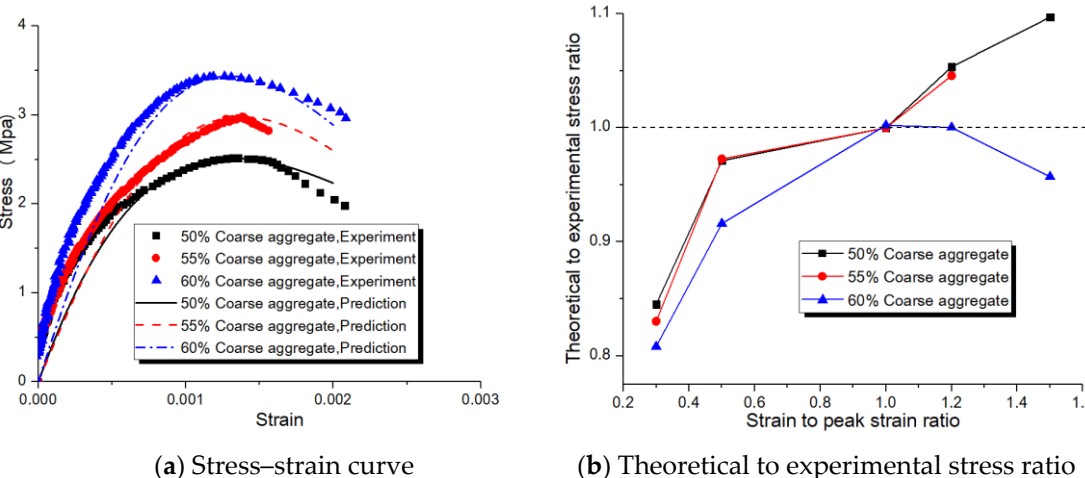

(**a**) Stress–strain curve                    (**b**) Theoretical to experimental stress ratio

**Figure 15.** (**a**) Theoretical and experimental compressive stress–strain curve and (**b**) theoretical to experimental stress ratio of CBA RCC with 9% cement.

Figure 16 presents the relationship between damage variable *D* and the compressive strain of CBA RCC. Figure 16a demonstrates that the increase in coarse aggregate from 50 to 55 and finally to 60% effectively delays the generation of damage and inhibits the propagation of damage before peak stress. However, after peak stress, the damage variables *D* tend to be close to each other. This indicates that the coarse aggregate plays its interlock effect mainly before peak stress, the interlock effect of coarse aggregate is lost when the aggregate is pulled out from matrix. When the coarse aggregate content was 50%, the increase in compacting load resulted in the increase in the damage speed of RCC before peak stress, because the high content of highly porous CBA was easy to be damaged under higher compacting load. In contrast, when the coarse aggregate content was 60%, the increase in compacting load decreased the damage speed of RCC in the entire process, because the high content of coarse aggregate was easy to be subjected to more compact under higher compacting load.

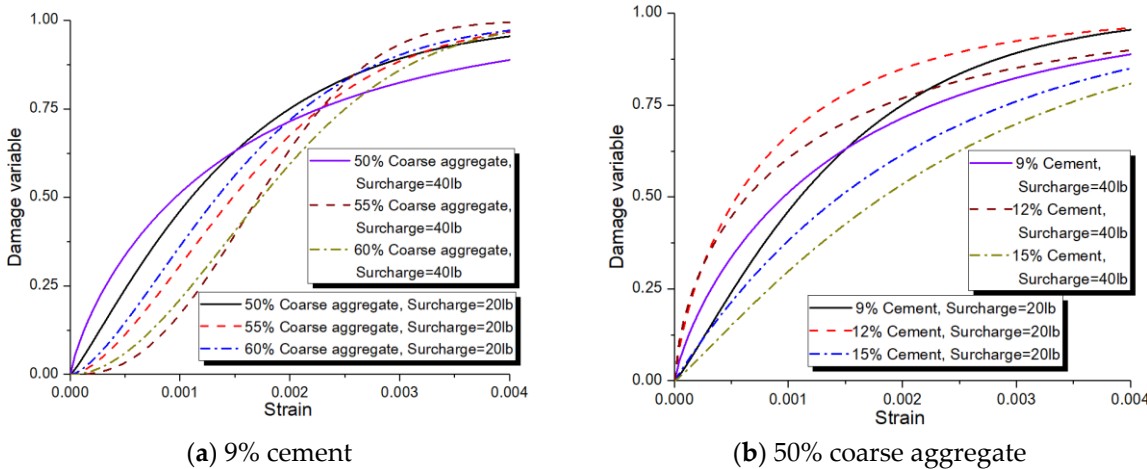

(**a**) 9% cement                    (**b**) 50% coarse aggregate

**Figure 16.** Damage variable vs. compressive strain of RCC with (**a**) 9% cement and (**b**) 50% coarse aggregate.

Figure 16b illustrates that the increase in cement content from 9 to 12% could not inhibit the propagation of damage and even increased the damage variable before peak stress. Nevertheless, the increase in cement content from 12 to 15% effectively delayed the generation of damage and inhibited the propagation of damage in the entire loading process. This indicates that adequate cement is necessary to improve the damage resistance of concrete, which is necessary to bond the solid particles together. In general, the increase in compacting load significantly decreased the damage speed of RCC in the entire process. When cement content was 15% and compacting load was 40 lb, the CBA RCC reached the lowest damage speed. Therefore, enough cement is also helpful for the compacting load to play its role, because the high content of cement can bond the solids more compactly under higher compacting load.

At the same time, in order to verify the accuracy of the model herein, the theoretical stress calculated by using the model in this study was compared with the experimental stress value mentioned in the literature report [38]. The comparison results are presented in Table 5. The theoretical formula can accurately illustrate the stress–strain curves of RCC reported in the literature, most of the theoretical to experimental stress ratios are in the range from 0.96 to 1.00.

**Table 5.** Comparison between theoretical and experimental stress of RCC from literature report.

| Strain to Peak Strain Ratio | 0.3 | 0.5 | 1 | 1.2 | 1.5 |
|---|---|---|---|---|---|
| Experimental Stress | 13.44262295 | 17.29964 | 20.72404 | 20.27322 | 19.05738 |
| Theoretical Stress | 11.94926377 | 16.70958 | 20.70002 | 20.28297 | 18.54175 |
| Theoretical to Experimental Stress Ratio | 0.888908646 | 0.965892 | 0.998841 | 1.000481 | 0.972943 |

## 4. Conclusions

Based on the experimental investigation of uniaxial compressive and splitting tensile behavior of CBA RCC, the uniaxial compressive damage constitutive model was established and the following conclusions are drawn:

(1) Variation in compaction load showed no strong and distinct influence on compressive strength, elasticity modulus, and splitting tensile strength of CBA RCC at 28 days in this experimental program. Nevertheless, higher compaction load increased the toughness and deformability of CBA RCC.

(2) The compressive strength and the splitting tensile strength of CBA RCC were found to be greatly improved with the increase in cement contents. Varying cement content showed more effect on compressive strength than that on splitting tensile strength. The relationship between compressive and splitting tensile strength was proposed, which is very close to the relationship recommended by the CEB-FIP model code.

(3) The compressive strength, elasticity modulus, and splitting tensile strength decreased with the increased CBA content.

(4) The uniaxial compressive stress–strain constitutive model of CBA RCC was established. The uniaxial compressive constitutive model based on the continuum damage theory could well illustrate the stress–strain relationship of CBA RCC. The growth process of the damage variable demonstrates the hybrid effect of coarse aggregate, cement, and compacting load on delaying damage under uniaxial compression. The theoretical formula can also accurately illustrate the stress–strain curves of RCC in literature.

**Author Contributions:** Conceptualization, Y.L. and V.B.; methodology, Y.L. and V.B.; validation, Y.L., L.L. and V.B.; formal analysis, Y.L.; investigation, Y.L.; resources, Y.L.; data curation, Y.L.; writing—original draft preparation, L.L.; writing—review and editing, L.L.; visualization, L.L.; supervision, Y.L.; funding acquisition, L.L. All authors have read and agreed to the published version of the manuscript.

**Funding:** This research was funded by the Natural Science Basic Research Plan in Shaanxi Province of China (Program No. 2021JQ-174), Fundamental Research Funds for the Central Universities (2452020054), and the Opening Project of State Key Laboratory of Green Building Materials (2020GBM10). The APC was funded by Fundamental Research Funds for the Central Universities (2452020054) and the Opening Project of State Key Laboratory of Green Building Materials (2020GBM10).

**Institutional Review Board Statement:** Not applicable.

**Informed Consent Statement:** Not applicable.

**Data Availability Statement:** Data are available on request from the authors.

**Acknowledgments:** The authors acknowledge the help from Annie Hole/Ted Hole Memorial Laboratory in the University of Alberta.

**Conflicts of Interest:** The authors declare no conflict of interest.

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
