# Peer review of "Constitutive Model of Uniaxial Compressive Behavior for Roller-Compacted Concrete Using Coal Bottom Ash Entirely as Fine Aggregate"

_buildings, doi:10.3390/buildings11050191_

Round 1
Reviewer 1 Report
Manuscript Number: buildings-1177683
Title: Constitutive model of uniaxial compressive behavior for roller compacted concrete using coal bottom ash entirely as fine aggregate
Statut : Accepted with minor corrections
The paper is well written, well-structured and well detailed. However, I have some remarks and corrections to propose
Justify the choice of the proportions of Bottom Ash in the mixtures studied.
Figure 1 and 2 are not necessary, they can be removed.
For all the figures, remove the gray background and enlarge them because they are invisible. Enter the confidence intervals for the values, because the test was carried out on several specimens.
Fig 4. For a better comparison, scale the Rc of figs: a, b, and c to 19MPa.
Fig 7. For a better comparison set the Y axis scale of the 3 figures (a, b, c) to 3MPa.
Fig 10. For a better comparison set the Y axis scale of the 3 figures (a, b, c) to 0.2.
Fig 14. For a better comparison set the Y axis scale of the 3 figures (a, b, c) to 18 GPa.
Fig 16. For a better comparison, scale the Y axis of the 2 figures (a, b) to 9 MPa. And correct (MPa and not Mpa).
Reviewer 2 Report
Reviewers' comments:
Manuscript ID: buildings-1177683
Title: Constitutive model of uniaxial compressive behavior for roller compacted concrete using coal bottom ash entirely as fine aggregate.
Manuscript Type: Article.
Reviewers' comments:
The manuscript describes the Constitutive model of uniaxial compressive behavior for roller compacted concrete using coal bottom ash entirely as fine aggregate. The manuscript needs a detailed editing. Some markings are made to just illustrate the extent of editing needed. A thorough revision addressing all the concerns is needed and if the authors are prepared to do that it can be considered for a review of the revised manuscript.
The authors need to consider the following comments
- In the Abstract, the authors need to improve with more specific short results and conclusions, i.e. academic novelty or technical advantages.
- Introduction part is lagging the previous studies on. Literature review is not updated in the manuscript. More recent literature need to be added.
- Figures 3, 5, and 6 - is not clear make clear.
- Figures 8, 9, and 11 - is not clear make clear.
- The words in conclusion are too more; and they need to be simplified.
- Several faults: are added or missing spaces between words: see PDF file.
- References: author should use order and there are recent references in 2020-2021 treating the same subject, you can use.
- Make all references in same format for volume number, page number and journal name, because it is difficult to searching and reading.
- Furthermore, they should add the graphical abstract, it is use full to readers.
- Minor English corrections is required throughout the manuscript.
Author Response
Manuscript ID: buildings-1177683
Type of manuscript: Article
Title: Constitutive model of uniaxial compressive behavior for roller compacted concrete using coal bottom ash entirely as fine aggregate
Journal: Buildings
Dear editor and reviewers,
I am very grateful to you for your recommendations. The paper has been revised with all amendments and all forms have been corrected according to the editor's requirements and author guidelines. All revisions have been clearly highlighted, using the "Track Changes" function in Microsoft Word. As well, Please find attached here with the Answers to reviewers' questions. Your acknowledging the receipt of the manuscript would be highly appreciated.
Thank you and best regards
Please see the attachment.

Reviewer 3 Report
Some remarks to Authors:
Introduction part is not well written in terms of scientific format, In lines 49-55, very poor English, an also lines 66-74. In general, it is hard to understand if authors talk about their work or literature work. Almost no discussion about ash, only river sand issue. I believe using coal ash has more positive impact on the nature by decreasing the negative affects of land filled ashes, not only less usage of river sand. In general, number of citations are also very limited in the entire manuscript. No discussion about the model based studies as well, which should be the main topic of the work.
Authors should add chemical and physical characterization of ash they used. Otherwise there is no point to make any comment on the results. After seeing such analysis, reviewing process should continue, at the current stage there is not much to discuss.
Author Response
Manuscript ID: buildings-1177683
Type of manuscript: Article
Title: Constitutive model of uniaxial compressive behavior for roller compacted concrete using coal bottom ash entirely as fine aggregate
Journal: Buildings
Dear editor and reviewers,
I am very grateful to you for your recommendations. The paper has been revised with all amendments and all forms have been corrected according to the editor's requirements and author guidelines. All revisions have been clearly highlighted, using the "Track Changes" function in Microsoft Word. As well, Please find attached here with the Answers to reviewers' questions. Your acknowledging the receipt of the manuscript would be highly appreciated.
Thank you and best regards
Answers to reviewers' questions
Reviewer #3: Introduction part is not well written in terms of scientific format,
Answer:
Thank you for your suggestion. The authors have improved the abstract with more specific short results and conclusions, i.e. academic novelty or technical advantages. It is on Page 1-2, see it as follows.
In lines 49-55, very poor English, an also lines 66-74. In general, it is hard to understand if authors talk about their work or literature work. Almost no discussion about ash, only river sand issue. I believe using coal ash has more positive impact on the nature by decreasing the negative affects of land filled ashes, not only less usage of river sand.
Answer:
Thank you for your suggestion. The English in lines 49-55 and lines 66-74 have been improved. More discussions about ash have been added to address the positive impact of coal ash. It is on Page 4-5, see it as attachment.
In general, number of citations are also very limited in the entire manuscript.
Answer:
Thank you for your suggestion. Number of citations are increased from 17 to 38 in the entire manuscript, see it as attachment.
No discussion about the model based studies as well, which should be the main topic of the work.
Answer:
Thank you for your suggestion. More discussion about the model based studies have been added, see it as attachment.
Authors should add chemical and physical characterization of ash they used. Otherwise there is no point to make any comment on the results. After seeing such analysis, reviewing process should continue, at the current stage there is not much to discuss.
Answer:
Thank you for your suggestion. The chemical and physical characterization of ash have been added in section 2.1 (Page 6), see it as follows.The maximum size, density, water absorption, and fineness modulus of CBA are 5.00 mm, 1.99 g/cm3, 10.0%, and 3.01, respectively. The X Ray Fluorescence test presents that CBA mainly consist of SiO2 (49.9 %), Al2O3 (13.1 %) and Fe2O3 (23.0 %), accompanied by little CaO (0.80 %), MgO (0.38 %), SO3 (0.26 %) and so on.

Round 2
Reviewer 2 Report
Reviewers' comments:
The authors revised the manuscript according to the reviewers' comments.
So that I recommended this manuscript accept for publication in Buildings.
Author Response
The reviewer said,"The authors revised the manuscript according to the reviewers' comments. So that I recommended this manuscript accept for publication in Buildings." So I think it is OK in this form.
Reviewer 3 Report
Thank you for re-working on the paper. There are still few issues (mainly language related). It is better, if authors benefit from a professional language editing service as it can improve the quality of paper.
For example; 1- there are sentences starting with "but, so" should be rephrased
2- (line 113) using "so on" is not a good way of explaining the oxides in your coal ash.
3- There are grammatical case errors in the entire text, usage of past tense in the results should not be very common.
